# Improving Adherence to a Home Rehabilitation Plan for Chronic Neck Pain through Immersive Virtual Reality: A Case Report

**DOI:** 10.3390/jcm12051926

**Published:** 2023-02-28

**Authors:** Matteo Cioeta, Sanaz Pournajaf, Michela Goffredo, Giuseppe Giovannico, Marco Franceschini

**Affiliations:** 1Department of Neurological and Rehabilitation Sciences, IRCCS San Raffaele Roma, 000163 Rome, Italy; 2Department of Medicine and Health Science “Vincenzo Tiberio”, University of Molise, 86100 Campobasso, Italy; 3Department of Human Sciences and Promotion of the Quality of Life, San Raffaele University, 00166 Rome, Italy

**Keywords:** virtual reality, chronic pain, neck pain, patient-centered care, treatment adherence and compliance

## Abstract

Idiopathic chronic neck pain is a highly disabling musculoskeletal condition. Immersive virtual reality shows a promising efficacy in the treatment of chronic cervical pain through the mechanism of distraction from the pain. This case report describes the management of C.F., a fifty-seven-year-old woman, who suffered from neck pain for fifteen months. She had already undergone a cycle of physiotherapy treatments including education, manual therapy, and exercises, following international guidelines. The patient’s poor compliance did not allow adherence to the exercise’s prescription. Home exercise training through virtual reality was therefore proposed to the patient to improve her adherence to the treatment plan. The personalization of the treatment allowed the patient to resolve in a short time period her problem and return to live with her family peacefully.

## 1. Introduction

Chronic neck pain is a widespread musculoskeletal condition highly prevalent in the population of workers causing high levels of disability, anxiety, and stress; repetitive work and low social support at work are risk factors for this condition [1,2,3]. Musculoskeletal pain has a multidimensional origin, as stated by the ultimate definition conjured by the International Association for the Study of Pain: pain is always a personal experience that is influenced to varying degrees by biological, psychological, and social factors. [4]. The identification of psychological factors is important in order to allow for a personalized treatment program and better outcomes [5]. Kinesiophobia is defined as an excessive, irrational, and debilitating fear of moving due to a feeling of vulnerability to injury or re-injury [6]. This is a common condition in patients with chronic neck pain [7]. Pool et al. showed that the “fear of movement” could prevent a full recovery in patients with sub-acute chronic neck pain, and Asiri et al. demonstrated how kinesiophobia is correlated to neck pain intensity and a significant predictor of pain intensity, proprioception impairments, and functional performance [8,9,10]. Physiotherapists have different ways of dealing with this issue: education, manual therapy, and therapeutic exercise are the gold standards to treat chronic neck pain [11,12]. Moreover, in recent years, treatments with psychological approaches have emerged in the musculoskeletal field, and the results are promising [13]. Thus, developing other approaches to target more than the biological structure requires the work of further studies. Virtual reality has proven to be a valid tool that can help physiotherapists in managing patients with different musculoskeletal and rheumatological conditions, from rheumatoid arthritis and fibromyalgia to ankle instability and total knee replacement [14,15,16]. The outcomes included pain, quality of life, and balance, but in the majority of the studies, a non-immersive virtual reality was used. Furthermore, new evidence of immersive virtual reality indicates positive results in patients with chronic neck pain [17,18]; in particular, Tejera et al. discovered its effectiveness in reducing kinesiophobia at the three-month follow-up [18]. In the context of the biopsychosocial management of musculoskeletal disorders, an adherence to exercise prescription is another aspect to be addressed. Adherence is defined as “the extent to which the patient undertakes the clinic-based and home-based prescribed components of the physiotherapy program” [19]. It is necessary that physiotherapists can use all the tools validated in the literature in order to individualize the most appropriate treatment path for the patient and improve treatment adherence [20].

This case report provides an example of clinical reasoning that leads to the management of a patient with chronic neck pain through an innovative home treatment tool with the aim of improving adherence to the exercise program.

## 2. Methods

This case report follows the CARE guidelines [21].

## 3. Case Presentation

### 3.1. Past and Present Medical History

C.F. is a 57-year-old woman of white ethnicity. She is married, has two children, and is a doctor of laboratory analysis. She arrived at our private clinic on 15 November 2021 for neck pain. The patient reported the onset of this disorder about 15 months earlier; after a very tiring day at work, she began to experience severe pain in the right side of the neck (Figure 1). She had suffered from a few episodes of neck pain in the past, all of which resolved within a few days by taking non-steroidal anti-inflammatory drugs (NSAIDs). She reported pain of 5/10 on the Numerical Pain Rating Scale (NPRS) [22]. She was worried about her condition (“this time it’s different from all the others; after all this time I still feel a lot of pain and I can’t move as before, I’m worried and I’m afraid. I won’t solve this problem”). She went to the general practitioner a week after the onset because the NSAIDs had no effect; she was prescribed a course of oral cortisone. Successively, she conducted ten sessions of physical therapy without receiving any results; the patient explained that the approach proposed by the physiotherapist was based on education, manual therapy techniques, and exercises to be done both in session and at home. In recent months, due to the pandemic, she has been forced into very stressful shifts at the hospital and a lack of staff on the ward did not allow her adequate rest; she often worked in a sitting position, both in front of a microscope and in front of a PC, and her condition did not allow her to work at her best. All of this also affects her family life as she has little time to spend with them. In anamnesis, she did not report symptoms related to a non-musculoskeletal disease [23]. She has been assumed drugs for hypertension for about ten years. We then decided to continue with the objective examination.

### 3.2. Objective Exam

The physical examination of the patient began with an observation of the anterior, posterior, and sagittal planes and excluded morphological and cutaneous alterations. On palpation, tenderness at the level of the upper trapezius, anterior, and middle right scalenus was observed. The active cervical range of motion (ROM) was painful (5/10 NPRS) and partially limited in all planes, especially in the right rotation (Table 1).

Considering the limitations described above in the active ROM and the tenderness found at palpation, we proceeded with the following passive segmental tests: the C0-C2 axial rotation test [24], posterior–anterior middle cervical glide test [25], and cervical rotation lateral flexion test [26]; these tests were negative and with full asymptomatic ROM. Taking into account these limitations, the patient was evaluated with active and passive movements in the supine position; the limitation to the active movement was the same as in the pain-free sitting position (0 NPRS). The strength of the cervical muscles was evaluated only in the supine position due to pain in the sitting position. Right and left rotation and right and left lateral flexion resulted in a four on the Medical Research Council (MRC) scale [27], whose maximum score is five. A deep neck flexor endurance test was performed, and it was 11 s, far from the normative data in the female population (29.3 ± 13.7 s) [28].

### 3.3. Diagnostic Evaluation

Given the information collected about the patient’s history, we proceed with the administration of the following patient-reported outcome measures (PROMs): the neck disability index (NDI) [29] with a score of 46 and the Tampa scale of kinesiophobia (TSK) [30] with a score of 48. This directs towards the definition of a patient with idiopathic chronic neck pain with high levels of kinesiophobia.

## 4. Timeline

Figure 2 depicts the timeline of the study. 

## 5. Intervention

Based on information collected in medical history, the clinical examination data and PROMS scores of the previous rehabilitation program were discussed with the patient. She reported that the physiotherapist explained her condition, the non-malignant origin of her pain, and the “pain alarm system”; he encouraged her to take an active approach to the condition and educated her to frequently change position at work. They also agreed to do an exercise program at home (including strength training of the neck and shoulder muscles). Unfortunately, the patient reported that she had not always performed the exercises proposed at home by the physiotherapist due to the lack of time and will; she did not like those exercises, and she considered them useless even though she understood all of the explanations. Home training with a virtual reality headset (Figure 3a,b) was thus proposed to her. The patient welcomed the proposal to carry out this type of treatment and started training that did not have any side effects.

The training consisted of the explanation of the treatment modality (VR Ocean Aquarium app, Figure 3a) and familiarization with the tool (Virtual Reality Headset, DESTEK, London, UK); it is a commercial headset that works with a large number of smartphones with a screen from 4.7 to 6.8 inches (Figure 3b). The patient was informed of the cost of the device (56.99 euros) so that she could purchase it for home rehabilitation. The game application (available for free on the Google Play Store, Google LLC) consisted of being immersed in a VR environment similar to water and observing different species of fish around; once the pointer encounters an animal, its name appears on the screen. The baseline dosage for the home program development was designed according to the “time-contingent approach” and graded activity [31]. In her medical history, the patient reported that she had worked several hours in a sitting position, so the patient was asked the minimum time necessary in which she had to remain in the same position (both under the microscope and on the computer) and how much she managed to maintain those positions; then, rest was chosen between repetition based on how she could rest between each change in position at work. After collecting this information, the home program was designed as reported in Table 2. The app was downloaded to the patient’s smartphone and a first follow-up was then set for the following week.

### 5.1. First Follow-Up

On 22 November 2021 (seven days after the start of the VR training), the patient reported enjoyment during the treatment and feeling better with “less heaviness in the neck and shoulder”. The active ROM, both in sitting and supine positions, was complete in all directions, with a residual deficit in the right rotation (55°) only. The NDI and TSK were administrated and scored 18 and 35, respectively, while the NPRS was 3/5. The deep neck flexor endurance test was performed and resulted in 18 s. The patient performed all the exercises and asked the physiotherapist to increase the training time. As a result of the engagement in the treatment plan and the improvement in the condition, the possibility of extending the interval between sessions was discussed with the patient. As a result, a new program was developed for the next two weeks (Table 3).

### 5.2. Second Follow-Up

The second follow-up took place on 7 December 2021 and the patient reported almost a complete resolution of their symptoms. The active ROM of the neck was complete in all directions, the NDI and TSK scored 0 and 16, respectively, and NPRS was 1/5. (PROMs improvements are reported in Figure 4). The deep neck flexor endurance test was performed, and the result was 26 s. The patient reported that she did the exercise program every day and enjoyed it with her family as well. She could work better under the microscope and sitting was no longer a problem. Therefore, the patient was advised to continue the program for three days a week of her choice, increasing the dosage of each session for the next four weeks (Table 4). Two telephone follow-ups were planned: six weeks and three months later.

### 5.3. Third and Fourth Follow-Ups

The last two follow-ups were carried out by telephone six weeks and three months after the start of treatment. The patient reported that she performed the exercises consistently, only skipping five sessions in three months. The patient reported that she had not had any episodes of neck pain and that she had started a postural gymnastics course two weeks before to stay active.

At the end of the treatment plan, we asked the patient to report her impression of her recovery: “I have to thank the physiotherapist who followed me because he was able to understand my needs regarding the timing of the treatment. I really appreciated the possibility to perform the exercises comfortably at home, able to manage work commitments. In addition, the game was more fun than the last home rehabilitation and I also involved my little daughter in carrying the exercises!”.

## 6. Discussion

To the best of our knowledge, this is the first case report describing the clinical reasoning that leads to the choice of immersive virtual reality as a therapeutic option in the treatment of a musculoskeletal disorder and that describes in detail the posology of treatment through virtual reality. A strength of this case report is the choice of the most appropriate management at that time for the patient’s problem and the ongoing discussion with her about her preferences [32]. In her medical history, she reported following physiotherapy treatment recommended by several guidelines [11,12], but this did not lead to an improvement in her condition. The rehabilitation treatment should also take into account other aspects such as the patient’s preferences [33,34]. The exercises were not always performed by the patient. Considering her preferences made it possible to orient the therapeutic approach in a different way. It was therefore decided to use virtual reality to increase patient compliance. Patients have a greater satisfaction with using immersive virtual reality in their home rehabilitation program than with other treatments [35], and this could lead to an improved adherence to the exercise program. Another strength of this study is the possibility for the patient to use the virtual reality headset at home, purchasing it given its low cost. In the literature, expensive headsets are often used in trials and their application is not possible in a private clinic. In this case, an economic headset was proposed to the patient for the home exercise program.

Finally, in a musculoskeletal study, it is important to describe in detail the home exercise program to allow for the definition of an optimal dosage for patient recovery [36]; this is even more important in virtual reality, an emerging field of rehabilitation [37].

This case report also has some limitations, in particular regarding the mechanism of the symptomatology resolution. Taking into account the multidimensional nature of pain [4], the patient’s improvement can be attributed to many factors. The distraction from pain is one of the mechanisms by which virtual reality works; several neuroimaging studies have revealed that specific areas of the brain are activated when pain decreases during virtual reality training [38,39]. Thus, it is necessary to clarify that the effect that the treatment has had on the patient does not concern only training with virtual reality, other factors could contribute to the improvement of a patient’s symptoms and disability [40]. Still little known as a treatment, virtual reality may have created high expectations for improvement on the part of the patient: this mechanism has also been seen to occur with other types of treatment that have not been suggested by the guidelines in the treatment of neck pain [41]. This, together with a better adherence to treatment, has allowed the patient to solve her problem. Finally, a limitation of this study is that a follow-up longer than three months was not done.

Immersive virtual reality can be an additional strategy to improve the quality of life of a patient suffering from chronic pain and affordable alternatives can be used in clinical practice for personalized home treatment, as described in this paper. Further studies are needed to validate immersive virtual reality as a tool for the treatment of chronic musculoskeletal pain, but more importantly, further studies are needed to show the clinical reasoning on the choice of such treatment and to describe in detail the appropriate dosage for each patient.

## 7. Conclusions

This case report does not aim to demonstrate the effectiveness of immersive virtual reality in the treatment of chronic neck pain but to emphasize the importance of personalizing treatment by considering the possible causes of the failure of the previous treatment.

## Figures and Tables

**Figure 1 jcm-12-01926-f001:**
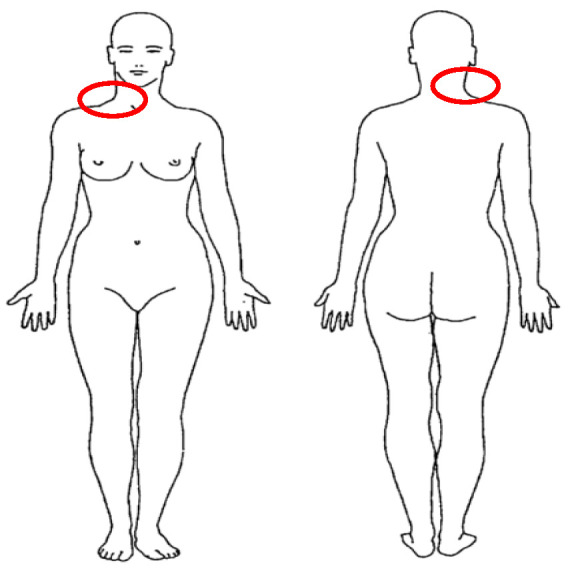
Body charts at the first evaluation.

**Figure 2 jcm-12-01926-f002:**
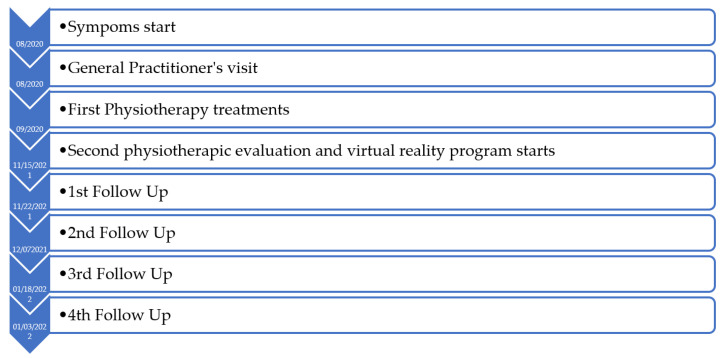
Timeline.

**Figure 3 jcm-12-01926-f003:**
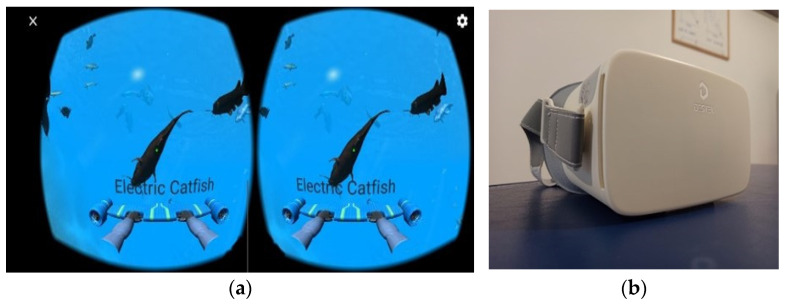
(**a**) VR Ocean Aquarium app. (**b**) Headset for home training (DESTEK Virtual Reality Headset).

**Figure 4 jcm-12-01926-f004:**
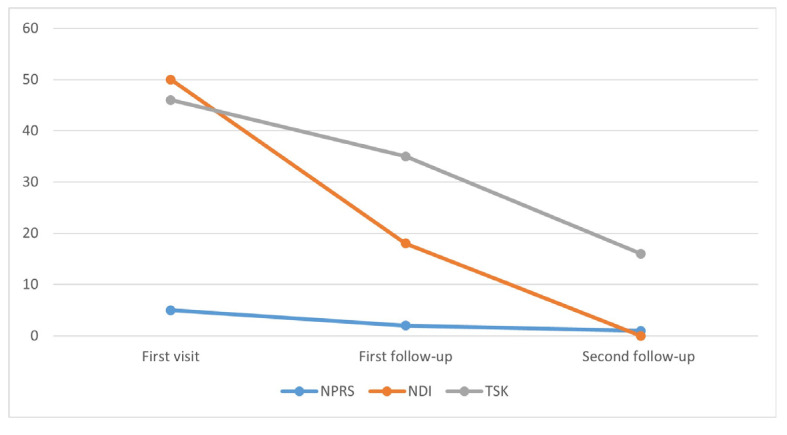
PROMs scores during the treatment—NPRS (numeric rating pain scale) scores from 0 to 10; NDI (neck disability index) scores from 0 to 50; TSK (Tampa scale of kinesiophobia) scores from 13 to 52.

**Table 1 jcm-12-01926-t001:** Measurement of active ROM (degrees) in a seated position.

Movement	Degrees (°)
Flexion	40
Extension	70
Right rotation	35
Left rotation	75
Right lateral flexion	35
Left lateral flexion	35

**Table 2 jcm-12-01926-t002:** Dosage of the virtual reality exercises for the first week of home treatment.

Day	Repetition (n°)	Repetition Length (s)	Rest between Repetition (s)
1	3	120	60
2	3	120	60
3	3	150	60
4	4	150	60
5	4	180	60

**Table 3 jcm-12-01926-t003:** Dosage of the virtual reality exercises for the second and third weeks of home treatment.

Day	Repetition (n°)	Repetition Length (s)	Rest between Repetition (s)
1	3	210	60
2	4	210	60
3	3	240	60
4	4	240	60
5	3	270	60

**Table 4 jcm-12-01926-t004:** Dosage of the virtual reality exercises for the following four weeks of home treatment.

Day	Repetition (n°)	Repetition Length (s)	Rest between Repetition (s)
1	4	270	60
2	4	300	60
3	4	330	60

## Data Availability

The data presented in this study are available on request from the corresponding author.

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
