# Peer review of "Improving Adherence to a Home Rehabilitation Plan for Chronic Neck Pain through Immersive Virtual Reality: A Case Report"

_jcm, 2023, doi:10.3390/jcm12051926_

Round 1

Reviewer 1 Report

Congratulations. I find it a very interesting article on a novel technique.

It needs some revisions to be published.

-The title needs to be revised. It should be more precise and formal.

-The abstract should include the objective of the study.

-The introduction should present more concisely the justification for the study.

-In the presentation of the case it would be convenient to explain in more detail the intervention process.

- It is necessary to add a section on the methodology used to prepare the article.

-The ethical aspects of the study should be indicated in more detail, indicating the code of ethics committee.

- The discussion aims to compare the results obtained with the rest of the literature published on the subject, sometimes this is not achieved. It needs to be longer. It would be advisable to add a section on the limitations of the study and future lines of research related to the subject.

-It is necessary to add a section on conclusions.

Author Response

Thank you for the revision of our paper. Please find the answers to your comments in the attached file.

Best regards

The authors

Reviewer 2 Report

Minor comments:

The title of the manuscript could be informative enough without it "More fun, less useless!"

Some spelling mistakes, for example on page 4, and you should check the consistency of the data in the graphic image (Timeline);

Is it possible to scientifically justify the choice of intervention intensity and repetitions in your study?

The discussion section mentions the strengths of the study but does not describe the limitations of the study

It would be good to highlight your conclusions with the main message at the end of the discussion

Author Response

(The authors gave the same response as above.)

Reviewer 3 Report

Main concerns:

1. At the beginning of the virtual reality program, ergonomic recommendations were made to limit time spend on microscope and at the computer, and to have rest breaks from this continuous work (Section 4. Intervention). These changes alone may be solely responsible for the improvement in neck pain - this needs to be added to discussion and limitations. There are also ergonomic clinical trials showing improved neck pain after appropriate recommendations - at least one relevant reference should be included.

2. During the assessment, 'passive segmental tests' led to 'complete and asymptomatic ROM' - Did the assessment procedure alleviate the neck pain?? This needs to be discussed in discussion and listed as a limitation.

3. No long-term follow-up after cessation of treatment is reported. This is a limitation.

*The first two listed concerns suggest provide enough evidence to suggest the virtual reality may not have had any effect on the neck pain. I suggest the authors soften the enthusiasm for the virtual reality and importantly add these serious limitations to the discussion and limitations,  then end with merely recommending that 'further research is necessary.' 

Author Response

(The authors gave the same response as above.)

Round 2

Reviewer 1 Report

The authors respond point by point to all my comments. The article is ready for publication.

Author Response

Thnak you for your comments.